# G Protein-Coupled Receptor 17 Inhibits Glucagon-like Peptide-1 Secretion via a Gi/o-Dependent Mechanism in Enteroendocrine Cells

**DOI:** 10.3390/biom15010009

**Published:** 2024-12-25

**Authors:** Jason M. Conley, Alexander Jochim, Carmella Evans-Molina, Val J. Watts, Hongxia Ren

**Affiliations:** 1Herman B Wells Center for Pediatric Research, Department of Pediatrics, Indiana University School of Medicine, Indianapolis, IN 46202, USA; jasmconl@iu.edu (J.M.C.); apjochim@uncg.edu (A.J.); cevansmo@iu.edu (C.E.-M.); 2Center for Diabetes and Metabolic Diseases, Indiana University School of Medicine, Indianapolis, IN 46202, USA; 3Department of Biochemistry & Molecular Biology, Indiana University School of Medicine, Indianapolis, IN 46202, USA; 4Roudebush VA Medical Center, Indianapolis, IN 46202, USA; 5Borch Department of Medicinal Chemistry and Molecular Pharmacology, Purdue University, West Lafayette, IN 47907, USA; wattsv@purdue.edu; 6Stark Neurosciences Research Institute, Indiana University School of Medicine, Indianapolis, IN 46202, USA; 7Department of Pharmacology & Toxicology, Indiana University School of Medicine, Indianapolis, IN 46202, USA; 8Department of Anatomy, Cell Biology & Physiology, Indiana University School of Medicine, Indianapolis, IN 46202, USA

**Keywords:** G protein-coupled receptor (GPCR), glucagon-like peptide 1 (GLP-1), signal transduction, cyclic AMP (cAMP), calcium, metabolic disease, diabetes, obesity

## Abstract

Gut peptides, including glucagon-like peptide-1 (GLP-1), regulate metabolic homeostasis and have emerged as the basis for multiple state-of-the-art diabetes and obesity therapies. We previously showed that G protein-coupled receptor 17 (GPR17) is expressed in intestinal enteroendocrine cells (EECs) and modulates nutrient-induced GLP-1 secretion. However, the GPR17-mediated molecular signaling pathways in EECs have yet to be fully deciphered. Here, we expressed the human GPR17 long isoform (hGPR17L) in GLUTag cells, a murine EEC line, and we used the GPR17 synthetic agonist MDL29,951 together with pharmacological probes and genetic approaches to quantitatively assess the contribution of GPR17 signaling to GLP-1 secretion. Constitutive hGPR17L activity inhibited GLP-1 secretion, and MDL29,951 treatment further inhibited this secretion, which was attenuated by treatment with the GPR17 antagonist HAMI3379. MDL29,951 promoted both Gi/o and Gq protein coupling to mediate cyclic AMP (cAMP) and calcium signaling. hGPR17L regulation of GLP-1 secretion appeared to be Gq-independent and dependent upon Gi/o signaling, but was not correlated with MDL29,951-induced whole-cell cAMP signaling. Our studies revealed key signaling mechanisms underlying the role of GPR17 in regulating GLP-1 secretion and suggest future opportunities for pharmacologically targeting GPR17 with inverse agonists to maximize GLP-1 secretion.

## 1. Introduction

Gut peptides exert important metabolic functions by promoting insulin secretion and regulating satiety [1,2]. Glucagon-like peptide-1 (GLP-1) is an incretin hormone secreted by enteroendocrine cells (EECs) in response to nutrient ingestion and is degraded rapidly by dipeptidylpeptidase 4 (DPP4) [3]. The physiological actions of GLP-1 are associated with regulation of appetite, food intake, and postprandial glucose excursions [3]. GLP-1 derived agonists and compounds that inhibit endogenous GLP-1 degradation are FDA-approved therapies for type 2 diabetes and obesity [4,5]. A potential alternative treatment strategy is through the modulation of endogenous GLP-1 secretion. It is well known that GLP-1 secretion from EECs is regulated by G protein-coupled receptors (GPCRs), but the exact signaling effectors and pathways are yet to be fully revealed.

GLP-1 secretion is influenced by G protein-coupling patterns and the balance of downstream second messenger signaling [6,7,8]. Gq-coupled receptor activation stimulates intracellular calcium signaling and increases GLP-1 secretion [9,10,11,12,13,14], and Gs-coupled receptor signaling leads to Gs-mediated activation of adenylyl cyclases, thereby increasing cAMP levels and GLP-1 secretion [15,16]. Direct modulation of cAMP levels also influences GLP-1 secretion. Specifically, increasing cellular cAMP levels by directly activating adenylyl cyclases with forskolin [17] and/or by inhibiting the breakdown of cAMP with phosphodiesterase inhibitors [17,18] stimulates GLP-1 secretion. Alternatively, inhibition of cAMP upon Gi-coupled receptor activation inhibits GLP-1 secretion [19,20]. As GPCRs are an important family of druggable proteins, understanding their precise biological actions in EECs will likely provide insight into developing new diabetes and obesity therapeutics by promoting endogenous GLP-1 secretion.

GPR17 was first reported as an orphan GPCR phylogenetically related to purinergic and cysteinyl-leukotriene (CysLT) receptors [21,22]. Our previous studies demonstrated the physiological role of hypothalamic *Gpr17* in maintaining metabolic homeostasis in rodents [23,24,25] and the potential implications of human *GPR17* genetic variants in metabolic diseases [26]. Moreover, we found that GPR17 is expressed in approximately 74% of GLP-1 expressing EECs of the human small intestine [27]. Acute inducible genetic ablation of *Gpr17* from mouse intestinal epithelium improves glucose homeostasis by enhancing nutrient-stimulated GLP-1 secretion and glucose-stimulated insulin secretion [27], suggesting that GPR17 may function to negatively regulate GLP-1 secretion. Targeting GPR17 to elevate GLP-1 secretion may represent another potential therapeutic approach for the treatment of metabolic diseases. GPR17 couples to multiple G proteins and regulates both cAMP and calcium signaling [26,28,29,30,31,32,33,34]. However, a clear understanding of GPR17 downstream signaling activities and the resulting physiological functions in intestinal EECs has yet to be established.

The present study was designed to investigate the GPR17 signaling mechanisms that regulate GLP-1 secretion. The native physiological function of GPR17 in EECs was modeled by constitutive hGPR17L signaling and regulation of GLP-1 secretion in GLUTag cells. A synthetic agonist, MDL29,951, was utilized as a small molecule tool to understand the contributions of GPR17 signaling pathways to the regulation of GLP-1 secretion. GPR17 signaling and GLP-1 secretion patterns were evaluated in combination with genetic approaches and signaling pathway inhibitors. Our study revealed underlying mechanisms for GPR17 regulation of GLP-1 secretion and provided insight to guide novel targeting strategies to maximize GLP-1 secretion as a treatment for metabolic diseases.

## 2. Materials and Methods

### 2.1. Materials

Forskolin (Cat. no. 1099), prostaglandin E_2_ (PGE_2_; Cat. no. 2296), and phorbol 12-myristate 13-acetate (PMA; Cat. no. 1201) were obtained from Tocris Bioscience (Minneapolis, MN, USA). HAMI3379 was obtained from Cayman Chemical (Ann Arbor, MI, USA; Item no. 10580). MDL29,951 was synthesized at Columbia University. 3-isobutyl-1-methylxanthine (IBMX; I5879), somatostatin (SST; S1763), pertussis toxin (PTX; P2980), and GPBAR-A (SML1207) were obtained from Millipore Sigma (St. Louis, MO, USA). YM-254890 was obtained from FUJIFILM Wako Pure Chemical Corporation (Richmond, VA, USA; 257-00631). Adenovirus vectors were constructed for Ad-EF1α-mCherry, Ad-CMV-HA-hGPR17L-EF1α-mCherry, Ad-CMV-HA-hGPR17L-V96M-EF1α-mCherry, and Ad-CMV-HA-mGPR17-EF1α-mCherry, and corresponding adenoviruses were generated and purified at Welgen, Inc. (Worcester, MA, USA). Plasmids encoding HA-tagged GPR17 (pcDNA3-HA-hGPR17L, pcDNA3-HA-hGPR17S, and pcDNA3-HA-mGPR17) were constructed by Welgen, Inc. (Worcester, MA, USA). The Luminescent cAMP biosensor pGloSensor-cAMP-22F was from Promega (Madison, WI, USA). The ArC-P2A-MeN plasmid that encodes a luminescent membrane β-arrestin recruitment reporter was generously provided by Dr. Jonathan A. Javitch (Columbia University) [35]. GLUTag cells were made available by Dr. Daniel J. Drucker (University of Toronto) and supplied by Dr. Sonia M. Najjar (Ohio University). HEK293 cells were obtained from ATCC (CRL-1573).

### 2.2. Cell Culture and Maintenance

GLUTag cells were maintained in Dulbecco’s Modified Eagle Medium (DMEM) supplemented with 10% fetal bovine serum and 1% penicillin–streptomycin in a humidified incubator at 37 °C and 5% CO_2_. For adenovirus transduction, GLUTag cells were seeded into poly-D-lysine coated 96-well plates or 4-chamber slides. The next day, cells were transduced with adenoviruses (100–1250 viral particles per cell at time of seeding) as indicated. After 24–30 h, adenovirus-containing media was changed to serum-free growth media for overnight serum starvation. The following day, GLUTag cells were used for cAMP or calcium signaling assays, GLP-1 secretion assays, or immunostaining and imaging.

HEK293 cells were maintained and transfected as previously described [26].

### 2.3. Cyclic AMP Measurements

Endpoint cAMP measurements were conducted as follows. GLUTag cells were seeded into poly-D-lysine-coated 96-well plates and transduced with adenovirus as specified above and then tested for cAMP signaling. Medium was removed and cells were washed with cAMP stimulation buffer (Hanks’ balanced salt solution (HBSS), 20 mM HEPES, pH 7.4) and then incubated in cAMP stimulation buffer for 30 min at 37 °C and 5% CO_2_. Cells were then treated for 30 min with agonist and forskolin in the presence of 500 µM IBMX at 37 °C and 5% CO_2_. Stimulation buffer was then decanted, and cAMP lysis buffer (50 mM HEPES, 10 mM CaCl_2_, 0.35% Triton X-100, and 500 µM IBMX) was added and shaken at 700 rpm for 1 h. Lysate was transferred to a 384-well plate, and the cAMP levels were quantified using the Cisbio HTRF cAMP (Gi kit; Revvity, Waltham, MA, USA) according to the manufacturer’s protocol.

Live-cell, real-time cAMP measurements were performed using Promega (Madison, WI, USA) GloSensor cAMP luminescent biosensor technology. Glosensor cAMP experiments were conducted as previously described in GLUTag cells [27] and HEK293 cells [26].

### 2.4. Calcium Measurements

GLUTag cells were seeded into black, clear bottom, poly-D-lysine-coated 96-well plates. Cells were transduced and serum starved as specified above. Growth medium was then replaced with 100 μL/well assay buffer (HBSS, 20 mM HEPES, pH 7.4) and 100 μL/well Calcium 6 loading dye (R8190; Molecular Devices, San Jose, CA, USA). During dye loading, cells were incubated at 37 °C and 5% CO_2_ for 2 h. MDL29,951 or KCl was added to the wells and fluorescence (485 nm excitation, 525 nm emission, and 515 nm automatic emission cutoff) was measured using a Molecular Devices FlexStation3 (Molecular Devices) at 37 °C. For experiments with PTX, cells were pretreated with 100 ng/mL PTX overnight and 100 ng/mL PTX was included with Calcium 6 loading dye for calcium flux assays. For experiments with YM-254890, cells were pretreated for 30 min at 37 °C.

Calcium flux in HEK293 cells was measured as previously described [26].

### 2.5. Membrane β-Arrestin Recruitment

Membrane β-arrestin recruitment measurements were conducted using a luminescent membrane β-arrestin recruitment reporter assay [35]. HEK293 cells were seeded into white, opaque, poly-D-lysine-coated 96-well plates and transiently transfected using Lipofectamine 3000 (Thermo Fisher Scientific, Waltham, MA, USA) according to the manufacturer’s protocol on the following day. At 48 h post-transfection, cells were washed twice with assay buffer (HBSS, 20 mM HEPES, pH 7.4) and allowed to equilibrate to room temperature for 15 min in assay buffer. Cells were then loaded with coelenterazine h (Nanolight Technology, Prolume Ltd., Lakeside, AZ, USA; 5 µM final concentration) for 5 min in the dark. Luminescence was read using a SpectraMax iD3 (Molecular Devices) with 1 s integration time. Baseline luminescence was read every 2 min for 6 min. Agonist was added and then luminescence was read every 2 min for 30 min.

### 2.6. GLP-1 Secretion and Quantification

GLUTag cells were seeded into poly-D-lysine-coated 96-well plates and transduced with adenovirus as specified above and then tested for GLP-1 secretion. Medium was removed and cells were washed two times with GLP-1 secretion buffer (138 mM NaCl, 4.5 mM KCl, 4.2 mM NaHCO_3_, 2.5 mM CaCl_2_, 1.2 mM MgCl_2_, 10 mM HEPES, 1 mM glucose, and 0.1% bovine serum albumin, pH 7.4). Cells were stimulated for 2 h with indicated treatments diluted in GLP-1 secretion buffer for a total volume of 200 µL/well at 37 °C and 5% CO_2_. Plates were then centrifuged at 300× *g* for 1 min at 4 °C, and 75 µL of the supernatant was removed and frozen at −80 °C until subsequent GLP-1 quantification. Total GLP-1 was quantified using the Meso Scale Discovery Total GLP-1 V-PLEX kit (Rockville, MD, USA; K1503PD) according to manufacturer’s instructions. For PTX treatment conditions, cells were pretreated with 100 ng/mL PTX overnight. For experiments with YM-254890, cells were pretreated for 15 min at 37 °C and 5% CO_2_.

### 2.7. Immunostaining and Imaging

GLUTag cells were seeded into poly-D-lysine-coated 4-chamber slides and transduced with adenoviruses the following day. To match GPR17 expression levels, cells were treated with 775, 1250, 200, and 100 viral particles per cell seeded for Ad-EF1α-mCherry, Ad-CMV-HA-hGPR17L-EF1α-mCherry, Ad-CMV-HA-hGPR17L-V96M-EF1α-mCherry, and Ad-CMV-HA-mGPR17-EF1α-mCherry, respectively. After 24–30 h, the transduction mix was replaced with serum-free growth media for overnight serum starvation. Cells were fixed, permeabilized, blocked, antibody stained, and counterstained with DAPI as described [26]. The following antibody staining conditions were used. Cells were incubated in primary antibody (1:1000 anti-HA.11 mouse, Biolegend, San Diego, CA, USA; clone 16B12) overnight at 4 °C and incubated in secondary antibody (1:500 DyLight 488 goat anti-mouse, Invitrogen, Thermo Fisher Scientific, Waltham, MA, USA; 35502) for 1 h at room temperature. After counterstaining, chambers were removed from the slides and coverslips were mounted with Aqua-Poly/Mount (Polysciences, Inc, Warrington, PA, USA). Slides were imaged for DAPI, Alexa488, and mCherry on a Leica Epifluorescence Adaptive Focus (Leica Microsystems Inc., Deerfield, IL, USA) microscope through a 40× objective lens.

### 2.8. Software and Data Analysis

Data were graphed and statistical analyses were performed using GraphPad Prism version 10 (GraphPad Software, Boston, MA, USA). Pharmacological measures from concentration–response curve analyses including pIC_50_ and pEC_50_ values were calculated using sigmoidal dose–response equations in GraphPad Prism.

## 3. Results

### 3.1. Human GPR17 Long Isoform Modulates cAMP Signaling in GLUTag Cells

Human GPR17 has two isoforms (i.e., long and short), and the long isoform differs from the short isoform by having an additional 28 amino acids at the extracellular N-terminus. The two isoforms display distinct expression patterns, with the human GPR17 long isoform (hGPR17L) more enriched in peripheral tissues [36]. To investigate the contributions of hGPR17L-mediated signaling in EECs, hGPR17L was expressed in GLP-1 secreting GLUTag cells [37,38]. GLUTag cells were transduced with adenoviruses encoding mCherry as a control or hGPR17L and mCherry under control of CMV and EF1α promoters, respectively. Cyclic AMP was measured following treatment with 10 μM forskolin (a list of compounds and molecular functions is provided in Table 1) and either vehicle or varying concentrations of MDL29,951 to activate hGPR17L.

Cells expressing hGPR17L had lower cAMP levels under forskolin stimulation conditions compared to control-transduced cells, suggesting constitutive activity. Furthermore, activation of hGPR17L with the agonist MDL29,951 resulted in concentration-dependent inhibition of forskolin-stimulated cAMP with a 29 nM IC_50_ (pIC_50_ = 7.54 ± 0.11) and maximum inhibition of 37 ± 5.1% at 390 nM MDL29,951. Similar to GPR17 cAMP modulation profiles in other cell types [26,30], reversal of inhibition was observed at higher concentrations of MDL29,951 (EC_50_ = 12 µM, pEC_50_ = 4.94 ± 0.09) in GLUTag cells expressing hGPR17L (Figure 1A).

The biphasic pattern of cAMP modulation was also observed upon activation of hGPR17L with MDL29,951 alone (Appendix A) or together with forskolin (Appendix A) when dynamic live-cell cAMP was measured with a bioluminescent cAMP biosensor. MDL29,951 had no effect on forskolin-stimulated cAMP in control-transduced cells, suggesting that the effects of MDL29,951 were mediated by hGPR17L expression in GLUTag cells (Figure 1A). Taken together, these results suggest that hGPR17L modulates cAMP signaling both constitutively and in response to agonist activation.

### 3.2. Human GPR17 Long Isoform Modulates Calcium Signaling in GLUTag Cells

GPR17 is also known to couple to Gq and modulate calcium flux [26,28,30,31,32]. Therefore, the effects of hGPR17L on calcium signaling in GLUTag cells were evaluated. MDL29,951 treatment led to concentration-dependent increases in calcium mobilization in GLUTag cells transduced with hGPR17L, but no effects were observed in control-transduced cells. Specifically, MDL29,951 stimulated calcium flux with 220 nM potency (pEC_50_ = 6.66 ± 0.13) and maximum peak ΔF/F of 0.80 ± 0.07 in cells expressing hGPR17L, whereas the peak ΔF/F in response to 30 µM MDL29,951 treatment was 0.01 ± 0.01 in control-transduced cells (Figure 1B). These results are consistent with hGPR17L-mediated calcium mobilization in response to agonist activation in GLUTag cells.

### 3.3. Human GPR17 Long Isoform Regulates cAMP and Calcium Signaling in GLUTag Cells Through Gi/o and Gq Signaling

The mechanisms contributing to the effects of hGPR17L on cAMP and calcium signaling were further investigated using pertussis toxin (PTX) to inhibit Gi/o-mediated signaling and YM-254890 to inhibit Gq signaling. PTX or YM-254890 treatment had no effects on basal cAMP (Figure 2A); however, PTX reversed the constitutive inhibition of forskolin-stimulated cAMP associated with hGPR17L expression (Figure 2A), suggesting that constitutive inhibition of cAMP by hGPR17L is mediated by Gi/o signaling. Furthermore, MDL29,951 was unable to inhibit forskolin-stimulated cAMP in the presence of PTX, but a stimulatory effect on cAMP was observed when cells were treated with >1 µM MDL29,951 (Figure 2B). In contrast, MDL29,951 inhibited forskolin-stimulated cAMP upon treatment with the Gq inhibitor YM-254890 (Figure 2B). Additionally, stronger inhibition of cAMP was observed at MDL29,951 concentrations ≥ 6.3 µM in conjunction with Gq inhibition (Figure 2B). These data suggest constitutive inhibition of cAMP upon expression of hGPR17L and that further inhibition upon agonist treatment is mediated by Gi/o signaling. The reversal of inhibition, or stimulatory cAMP effect, observed in response to MDL29,951 concentrations > 1 µM appears to be mediated (at least partially) by Gq signaling, consistent with GPR17 modulation of cAMP signaling in HEK293 cells [30,32]. The reversal of cAMP inhibition may be mediated by downstream Gq signaling that leads to activation of calcium/calmodulin- and/or protein kinase C-stimulated adenylyl cyclase isoforms [39]. Furthermore, higher concentrations of MDL29,951 are reported to stimulate Gs coupling [30], representing another possible mechanism for the reversal of cAMP inhibition. Future studies will be required to more fully understand these signaling mechanisms.

The contributions of Gi/o and Gq signaling to hGPR17L-mediated calcium mobilization were also tested. PTX treatment had no effect on the maximum calcium mobilization in response to MDL29,951 (97 ± 2.4% of the maximum response of the vehicle treatment condition) but was accompanied by a 3-fold shift in the EC_50_ of MDL29,951 from 440 nM (pEC_50_ = 6.34 ± 0.06) for vehicle-treated cells to 1300 nM (pEC_50_ = 5.89 ± 0.03) for PTX-treated cells, suggesting a role for Gi/o-mediated signaling (Figure 2C). Furthermore, YM-254890 treatment nearly abolished the MDL29,951-stimulated calcium response (17 ± 5.6% of the maximum response of the vehicle treatment condition), indicating that Gq signaling is necessary for calcium mobilization downstream of hGPR17L in GLUTag cells (Figure 2C). Notably, PTX and YM-25890 treatment had no effect on 15 mM KCl-stimulated calcium mobilization (Figure 2D), suggesting that the inhibitory effects observed on hGPR17L-mediated calcium signaling are not due to toxicity or non-specific effects on calcium signaling. In summary, the hGPR17L-mediated calcium mobilization in GLUTag cells appears to be dependent on both Gi/o and Gq signaling. These results are consistent with GPCR regulation of calcium signaling mediated by phospholipase C (PLC) [32,40]. Furthermore, GPR17-mediated calcium flux was previously reported to be dependent on Gβγ subunits (from Gi/o heterotrimeric G proteins) and Gαq stimulation of PLC [32].

### 3.4. Human GPR17 Long Isoform Negatively Regulates GLP-1 Secretion from GLUTag Cells

*Gpr17* knockout enhances nutrient-stimulated GLP-1 secretion in mice [27]. However, the effects of human GPR17 on GLP-1 secretion have yet to be investigated. GLUTag cells were transduced with adenoviruses encoding mCherry control or hGPR17L. GLP-1 secretion was measured in response to vehicle treatment or stimulation with 10 μM forskolin and 10 μM IBMX (F/I). As expected, F/I treatment stimulated GLP-1 secretion in both control- and hGPR17L-expressing cells; however, cells expressing hGPR17L had significantly reduced GLP-1 secretion under both vehicle and F/I-stimulated conditions (Figure 3A), suggesting that constitutive GPR17 activity inhibits GLP-1 secretion. F/I-stimulated cells were also treated with the GPR17 agonist MDL29,951. Surprisingly, low-dose MDL29,951 treatment (100 nM) had no effect on F/I-stimulated GLP-1 secretion in either control- or hGPR17L-transduced cells (Figure 3B). However, higher-dose MDL29,951 treatment (100 μM) significantly inhibited F/I-stimulated GLP-1 secretion in hGPR17L-expressing cells (i.e., 30 ± 7.7% inhibition) but not in control-transduced cells (Figure 3B). A similar pattern was observed in separate experiments in which the effects of hGPR17L expression and agonist activation on GLP-1 secretion were tested using varying concentrations of forskolin together with 10 μM IBMX. Cells expressing hGPR17L had lower GLP-1 secretion than control-transduced cells (Figure 3C). Additionally, 100 μM MDL29,951 (Figure 3C), but not 100 nM MDL29,951 (Appendix A), further inhibited GLP-1 secretion when GLUTag cells were stimulated with 1 μM forskolin and 10 μM IBMX. It is notable that the total magnitude of inhibition of GLP-1 secretion by hGPR17L (i.e., the combination of constitutive and agonist activation) was similar to the inhibition observed by somatostatin (SST), a known inhibitor of GLP-1 section [19], in control-transduced cells (Appendix A).

The pharmacological modulation of GLP-1 secretion by hGPR17L was evaluated more extensively through an MDL29,951 concentration–response analysis. MDL29,951 displayed concentration-dependent inhibition of F/I-stimulated GLP-1 secretion in GLUTag cells expressing hGPR17L with an IC_50_ of 300 nM (pIC_50_ = 6.53 ± 0.31) and maximum inhibition was observed for treatment with 25 μM MDL29,951 (Figure 3D). HAMI3379, a small molecule with GPR17 antagonist properties [41], significantly attenuated the MDL29,951-mediated inhibition of F/I-stimulated GLP-1 secretion, shifting the IC_50_ of MDL29,951 to 3600 nM (pIC_50_ = 5.45 ± 0.47) (Figure 3D). Such effects are consistent with the competitive antagonism reported for HAMI3379 [41], as the attenuation was not apparent in the presence of 100 µM MDL29,951. In summary, these data demonstrate that hGPR17L expression constitutively inhibited F/I-stimulated GLP-1 secretion and that MDL29,951 treatment further inhibited GLP-1 secretion specifically in GLUTag cells expressing hGPR17L. Furthermore, MDL29,951 inhibited F/I-stimulated GLP-1 secretion in a concentration-dependent manner that was attenuated by the GPR17 antagonist HAMI3379, suggesting that these effects on GLP-1 secretion were GPR17-mediated.

### 3.5. Human GPR17 Isoforms and Mouse GPR17 Have Distinct Second Messenger Signaling Profiles

GPR17-mediated signaling in GLUTag cells occurs through both Gi/o and Gq coupling, resulting in complex regulation of second messenger signaling via both cAMP and calcium. Therefore, functions of G protein signaling pathways in hGPR17L-mediated regulation of GLP-1 secretion were tested using genetic approaches and selective signaling pathway inhibitors.

The genetic approaches relied on GPR17 receptor forms that have altered signaling profiles relative to hGPR17L. We revealed here that mouse GPR17 (mGPR17) had distinct cAMP and calcium signaling relative to human GPR17. Human GPR17L, hGPR17S, and mGPR17 were expressed in HEK293 cells, and cAMP signaling modulation was measured following treatment with indicated concentrations of MDL29,951 and cAMP stimulation with forskolin (Appendix A). Concentration-dependent inhibition of forskolin-stimulated cAMP was observed for each GPR17 (IC_50_ values ranging from 1.5 to 3.3 nM), with maximal inhibition observed in the 100–300 nM MDL29,951 range (Appendix A). However, MDL29,951 concentration-dependent reversal of cAMP inhibition was observed for hGPR17L and hGPR17S, but this effect was less pronounced for mGPR17 (Appendix A). Furthermore, MDL29,951-stimulated calcium flux in HEK293 cells was similar for hGPR17L and hGPR17S (Appendix A), while cells expressing mGPR17 had an approximately 10-fold less potent calcium response to MDL29,951 and more than a 60% reduction in efficacy as compared to human GPR17 isoforms (Appendix A). MDL29,951-stimulated β-arrestin recruitment was similar for hGPR17L, hGPR17S, and mGPR17 (Appendix A). These data suggest that the balance of calcium and cAMP signaling may be distinct for human GPR17 and mouse GPR17 and may therefore differentially influence the regulation of GLP-1 secretion.

In addition to mGPR17, several naturally occurring hGPR17L variants have distinct downstream second-messenger signaling profiles when expressed in HEK293 cells [26]: human GPR17L-V96M has similar agonist-mediated inhibition of cAMP to that of hGPR17L-WT, but without the reversal of inhibition typically observed upon treatment with greater than 1 µM MDL29,951 [26], and hGPR17L-V96M shows impaired agonist-mediated calcium mobilization and β-arrestin recruitment [26]. Therefore, the hGPR17L-V96M signaling profile suggested that it could be used as a molecular tool to evaluate the role of distinct GPR17 signaling pathways in the regulation of GLP-1 secretion.

The distinct second-messenger signaling profiles observed for hGPR17L-V96M and mGPR17 relative to hGPR17L in HEK293 cells suggested that they could be used as molecular tools to evaluate the contributions of cAMP and calcium signaling to the regulation of GLP-1 secretion in GLUTag cells. Therefore, GLUTag cells were transduced with control, hGPR17L-WT, hGPR17L-V96M, and mGPR17 with multiplicities of infection that provided similar receptor expression levels as measured by immunofluorescence imaging using an α-HA antibody (Figure 4A). Efficient transduction was also confirmed by mCherry fluorescence imaging (Figure 4A). Downstream second messenger signaling and GLP-1 secretion were measured, and forskolin-stimulated cAMP responses were determined upon treatment with indicated concentrations of MDL29,951. GLUTag cells expressing mGPR17 demonstrated concentration-dependent inhibition of cAMP with maximal inhibition at 1 µM MDL29,951; however, MDL29,951 had no significant effect on forskolin-stimulated cAMP in cells expressing hGPR17L-V96M (Figure 4B). MDL29,951-stimulated calcium flux was also measured for cells expressing hGPR17L-V96M and mGPR17. Both receptor types stimulated calcium flux, but with less potency and efficacy as compared to hGPR17L-WT (Figure 4C). For example, MDL29,951 stimulated calcium mobilization in GLUTag cells expressing hGPR17L-V96M with an EC_50_ of 580 nM (pEC_50_ = 6.24 ± 0.07) and maximal stimulation that was 27 ± 1.5% of that observed for hGPR17L-WT (Figure 4C). Similarly, MDL29,951 stimulated calcium responses in GLUTag cells expressing mGPR17 with an EC_50_ of 680 nM (pEC_50_ = 6.17 ± 0.04), but with a maximal stimulation that was 74 ± 2.7% of the hGPR17L-WT response (Figure 4C). Notably, 100 µM MDL29,951 had no effect on F/I-stimulated GLP-1 secretion in GLUTag cells expressing hGPR17L-V96M, but significantly inhibited F/I-stimulated GLP-1 secretion in cells expressing mGPR17 (Figure 4D). Taken together, these data suggest that, when expressed in GLUTag cells, hGPR17L-V96M was unable to modulate cAMP signaling, had impaired calcium signaling function, and had no effect on GLP-1 secretion. Conversely, mGPR17 regulated cAMP signaling, calcium mobilization, and GLP-1 secretion similarly to hGPR17L-WT in GLUTag cells.

### 3.6. Human GPR17 Long Isoform Regulation of GLP-1 Secretion Is Dependent on Gi/o-Mediated Signaling in GLUTag Cells

Selective signaling pathway inhibitors were also used to dissect the GPR17 signaling pathways that contribute to GLP-1 secretion in GLUTag cells. PTX was used to test the role of hGPR17L-mediated Gi/o signaling on GLP-1 secretion. GLUTag cells were transduced with control or hGPR17L-encoding adenoviruses and the effect of 100 ng/mL PTX treatment was tested for modulation of GLP-1 secretion under basal, F/I-stimulated, and F/I + 100 µM MDL29,951 treatment. SST (1 µM) was included as a positive control for Gi/o-mediated effects on F/I-stimulated GLP-1 secretion. SST inhibited F/I-stimulated GLP-1 secretion, and this inhibition was attenuated with PTX (Appendix A), demonstrating that PTX is functioning to inhibit Gi/o signaling and regulation of GLP-1 secretion in the GLUTag cells. However, in control-transduced cells, PTX treatment had no effect on basal GLP-1 secretion or F/I-stimulated GLP-1 secretion, either alone or together with 100 µM MDL29,951 (Figure 5A). In contrast to control-transduced cells, GLUTag cells expressing hGPR17L tended to have relatively higher F/I-stimulated GLP-1 secretion in PTX-treated cells than matching control-treated cells (i.e., 99 ± 6.9 pM GLP-1 for PTX versus 83 ± 4.8 pM GLP-1 for control, *p* = 0.09 by unpaired t test) (Figure 5B and Appendix A). The F/I-stimulated GLP-1 secretion in hGPR17L-transduced cells was consistently higher in five of six experiments for PTX-treated as compared to control-treated (Appendix A), suggesting a likely role for Gi/o signaling in hGPR17L constitutive inhibition of F/I-stimulated GLP-1 secretion. Furthermore, PTX significantly attenuated 100 µM MDL29,951-mediated inhibition of F/I-stimulated GLP-1 secretion as compared to control-treated GLUTag cells expressing hGPR17L (Figure 5B). Taken together, these data suggest that Gi/o signaling pathways mediate both hGPR17L constitutive inhibition of GLP-1 secretion and GPR17 agonist-mediated inhibition of GLP-1 secretion.

### 3.7. Human GPR17 Long Isoform Regulation of GLP-1 Secretion Is Not Dependent on Gq-Mediated Signaling in GLUTag Cells

The Gq signaling inhibitor, YM-254890, was used to test the role of hGPR17L-mediated Gq signaling on GLP-1 secretion. The effect of 3 µM YM-254890 treatment was tested for modulation of GLP-1 secretion under basal, F/I-stimulated, or F/I + 100 µM MDL29,951 treatment conditions in control or hGPR17L-expressing GLUTag cells. YM-254890 treatment had strong inhibitory effects on F/I-stimulated GLP-1 secretion in both control and hGPR17L-expressing GLUTag cells (Appendix A). As such, the effect of MDL29,951 was expressed as a percentage of the F/I-stimulated GLP-1 secretion for each control and YM-254890 treatment conditions. As expected, MDL29,951 treatment had no effect on F/I-stimulated GLP-1 secretion in GLUTag cells transduced with control adenovirus (Appendix A), but inhibited F/I-stimulated GLP-1 secretion in cells expressing hGPR17L (Appendix A). The effects of MDL29,951 on F/I-stimulated GLP-1 secretion were similar for control and YM-254890 treatment conditions, suggesting that hGPR17L-mediated Gq signaling was not contributing to regulation of GLP-1 secretion (Appendix A).

The strong inhibitory effects of YM-254890 on F/I-stimulated GLP-1 secretion present a challenge for interpreting hGPR17L-mediated Gq signaling contributions to regulation of GLP-1 secretion. Furthermore, the effects may be due to off-target or non-specific inhibition. To protect against such off-target effects, it was of interest to define the lowest concentration of YM-254890 that maximally inhibits hGPR17L-mediated calcium flux. Therefore, a YM-254890 concentration-response analysis was conducted for inhibition of MDL29,951-stimulated calcium flux in GLUTag cells expressing hGPR17L. Concentration-dependent inhibition of MDL29,951-stimulated calcium mobilization was observed with robust inhibition at ~100 nM YM-254890 (Appendix A). In contrast, YM-254890 had no effect on 15 mM KCl-stimulated calcium flux in GLUTag cells expressing hGPR17L (Appendix A). These data suggest that concentrations as low as 100 nM YM-254890 were capable of inhibiting hGPR17L-mediated Gq signaling and that YM-254890 was likely not having toxic effects or non-specifically affecting calcium signaling.

In addition, 100 nM YM-254890 also inhibited F/I-stimulated GLP-1 secretion in GLUTag cells, but such robust inhibition was not observed when GLP-1 secretion was stimulated upon depolarization with 50 mM KCl or protein kinase C activation with 300 nM phorbol 12-myristate 13-acetate (PMA) (Appendix A). Moreover, a concentration–response evaluation suggested that YM-254890 had no effect on PMA-stimulated GLP-1 secretion, but inhibited F/I-stimulated GLP-1 secretion in a concentration-dependent manner (Appendix A). Subsequently, 100 nM YM-254890 treatment was tested for modulation of GLP-1 secretion under basal, PMA-stimulated, or PMA + 100 µM MDL29,951 treatment conditions in GLUTag cells transduced with control or hGPR17L-encoding adenoviruses. YM-254890 had no effect on PMA-stimulated GLP-1 secretion in either control (Appendix A) or hGPR17L (Appendix A)-expressing GLUTag cells. Additionally, YM-254890 had no effect on MDL29,951-mediated inhibition of PMA-stimulated GLP-1 secretion (Appendix A). In summary, the data suggest that hGPR17L-mediated Gq signaling was not contributing to the regulation of GLP-1 secretion.

### 3.8. Human GPR17 Long Isoform Distinctly Regulates GLP-1 Secretion and cAMP Signaling in a Manner That Is Dependent on Co-Stimulation Conditions in GLUTag Cells

F/I-mediated stimulation of GLP-1 secretion represents a robust means of stimulating GLP-1 secretion by artificial elevation of intracellular cAMP; however, EECs secrete GLP-1 in response to many physiological stimuli. Therefore, we proceeded to test the effects of hGPR17L expression and agonist stimulation on GLP-1 secretion in response to physiologically relevant modes of stimulation. Specifically, the effects of hGPR17L expression and agonist stimulation (100 nM MDL29,951 or 100 µM MDL29,951) on GLP-1 secretion were tested in GLUTag cells under basal conditions or when stimulated with Gs-coupled receptor agonists (e.g., bile acid receptor agonist GPBAR-A or prostaglandin receptor agonist PGE_2_) or protein kinase C stimulation (phorbol ester) conditions that are known to stimulate GLP-1 secretion (Figure 6A–D) [37,38,42,43]. Because hGPR17L signaling modulates cAMP signaling in a complex manner, the effects of hGPR17L expression and agonist stimulation on cAMP levels were also tested in GLUTag cells under the same treatment conditions (Figure 6E–H). Normalized data are presented to account for experiment-to-experiment variability and relatively smaller signal windows in some cases. The matching raw data graphs provide context relevant to the magnitude of such responses and are included for reference in Appendix A.

Under basal conditions, hGPR17L expression had no effect on GLP-1 secretion. Furthermore, MDL29,951 treatment had no effect on GLP-1 secretion in the control- or hGPR17L-transduced GLUTag cells (Figure 6A). Basal cAMP levels in GLUTag cells expressing hGPR17L were 91 ± 2.4% of the control-transduced cells (Figure 6E, *p* = 0.0631, one-sample t test compared to 100). Similarly, hGPR17L-expressing cells that were treated with 100 nM MDL29,951 had significantly lower cAMP than control-transduced cells, but significantly higher cAMP when treated with 100 µM MDL29,951 (Figure 6E). The relative cAMP increase was only observed in cells expressing hGPR17L (i.e., not in the control-transduced cells) and appeared to be agonist-concentration-dependent at concentrations greater than 1 µM MDL29,951 (Figure 6E).

Treatment with a GPBAR1 agonist, 10 µM GPBAR-A, led to a 2.67 ± 0.17-fold increase in GLP-1 secretion from GLUTag cells of the control transduction condition. Expression of hGPR17L significantly inhibited the GPBAR-A-stimulated GLP-1 secretion, consistent with constitutive inhibition of GLP-1 secretion (Figure 6B). Agonist treatment of either control- or hGPR17L-transduced cells had no statistically significant effect on GLP-1 secretion as compared to vehicle treatment (Figure 6B). However, it is notable that hGPR17L-expressing cells that were treated with 100 µM MDL29,951 had numerically lower GLP-1 secretion as compared to vehicle-treated cells (Figure 6B and Appendix A) and significantly lower secretion than vehicle-treated control cells (Figure 6B). These data suggest a combination effect of GPR17 constitutive and agonist-mediated inhibition of GPBAR-A-stimulated GLP-1 secretion. GLUTag cells expressing hGPR17L had significantly lower cAMP levels in response to 10 µM GPBAR-A stimulation than control-transduced cells (Figure 6F). GPR17 agonist treatment had no effect on GPBAR-A-stimulated cAMP levels in the control transduction condition, but there were modest MDL29,951 concentration-dependent increases in cAMP in cells expressing hGPR17L relative to vehicle treatment (Figure 6F). For example, GPBAR-A-stimulated cAMP levels were lower in cells expressing hGPR17L than control-transduced cells treated with 100 nM MDL29,951, but there was no significant difference observed when treated with 100 µM MDL29,951, as cAMP levels modestly increased in a concentration-dependent manner (Figure 6F).

Stimulation of prostaglandin receptors with 300 nM PGE_2_ yielded a 1.90 ± 0.06-fold increase in GLP-1 secretion from GLUTag cells of the control transduction condition (Figure 6C). Expression of hGPR17L significantly inhibited the PGE_2_-stimulated GLP-1 secretion, consistent with constitutive inhibition of GLP-1 secretion (Figure 6C). Agonist treatment of either control- or hGPR17L-transduced cells had no effect on PGE_2_-stimulated GLP-1 secretion as compared to vehicle treatment (Figure 6C). However, similar to effects observed when stimulated with GPBAR-A, hGPR17L expression and 100 µM MDL29,951 agonist activation significantly inhibited PGE_2_-stimulated GLP-1 secretion (Figure 6C and Appendix A). GLUTag cells expressing hGPR17L had cAMP levels in response to 300 nM PGE_2_ stimulation that were 83 ± 5.5% of the control-transduced cells (Figure 6G, *p* = 0.0856, one-sample *t* test compared to 100). MDL29,951 treatment had no effect on PGE_2_-stimulated cAMP levels in the control transduction condition, but agonist-concentration-dependent increases in cAMP were observed in cells expressing hGPR17L (Figure 6G). Similar to stimulation with GPBAR-A, PGE_2_-stimulated cAMP levels were lower in cells expressing hGPR17L than control-transduced cells treated with 100 nM MDL29,951. In contrast, hGPR17L-expressing cells had significantly higher PGE2-stimulated cAMP levels than control-transduced cells when treated with 100 µM MDL29,951 (Figure 6G).

Upon treatment with a PKC activator, 300 nM PMA, a 4.14 ± 0.18-fold increase in GLP-1 secretion was observed from GLUTag cells in the control transduction condition (Figure 6D). Expression of hGPR17L had no effect on PMA-stimulated GLP-1 secretion (Figure 6D). Furthermore, MDL29,951 treatment of either control- or hGPR17L-transduced cells had no significant effect on PMA-stimulated GLP-1 secretion as compared to vehicle treatment conditions (Figure 6D). However, although hGPR17L expression and agonist treatment had no effect on 300 nM PMA-stimulated GLP-1 secretion independently, together they significantly inhibited the stimulated GLP-1 secretion as compared to the control transduction condition without agonist (Figure 6D). Interestingly, the raw GLP-1 secretion data suggest that this effect may be driven by agonist activation, as 100 µM MDL29,951 treatment significantly inhibited PMA-stimulated GLP-1 secretion as compared to vehicle-treated GLUTag cells expressing hGPR17L (Appendix A). No significant differences were observed in cAMP levels between control- and hGPR17L-transduced GLUTag cells when treated with 300 nM PMA in the absence of agonist or with 100 nM MDL29,951 (Figure 6H). Although MDL29,951 had no effect on cAMP levels upon PMA treatment of control-transduced cells, GPR17 agonist-concentration-dependent increases in cAMP were observed in cells expressing hGPR17L (Figure 6H). Such effects were evident with the 100 µM MDL29,951 treatment condition, as the PMA-stimulated cAMP levels were significantly higher in the hGPR17L-transduced cells as compared to the control-transduced cells (Figure 6H).

## 4. Discussion

GLP-1 is an incretin hormone secreted by EECs in response to nutrients and metabolites, many of which are ligands for GPCRs expressed in EECs [3,44]. Understanding the cellular mechanisms and signaling pathways regulating endogenous GLP-1 secretion may provide alternative therapeutic targets for diabetes and obesity treatment. Our recent study reported that acute inducible genetic ablation of *Gpr17* in the mouse intestinal epithelium enhances nutrient-stimulated GLP-1 secretion and that GPR17 is expressed in GLP-1-expressing cells of the human gastrointestinal tract [27]. However, the functional effects of human GPR17 on GLP-1 secretion were not evaluated. Furthermore, GPR17 signaling studies have heavily relied on cell models with limited physiological relevance (e.g., HEK293 cells) and these studies indicate complex G protein-coupling patterns and second-messenger signaling profiles [26,27,28,30,31,32,34]. Here, we used a physiologically relevant intestinal enteroendocrine cell line (i.e., GLUTag cells) and directly assessed the contribution of GPR17 downstream signaling pathways to GLP-1 secretion. Human GPR17L was capable of modulating cAMP and calcium signaling in a complex manner that was mediated by coupling to Gi/o and Gq. The rise of cellular calcium is often associated with the hormone secretion from endocrine cells. For instance, Gq-coupled GPCRs including FFAR1 (GPR40) [9,10,11,14,45] and calcium-sensing receptor (CaSR) [12,13] stimulate increases in GLP-1 secretion. However, our data suggest that hGPR17L Gq-mediated signaling had no effects on GLP-1 secretion. In addition to efforts to reveal the rules and mechanisms governing the G protein-coupling selectivity of GPCRs [46], our studies highlight the importance of establishing the functional relevance of G protein coupling in each physiological setting.

The dynamic modulation of cellular cAMP with Gi/o- and Gs-coupled GPCR signaling and cAMP modulators often predicts hormonal secretion in EECs [6,7,8]. For example, elevation of cellular cAMP levels by Gs-coupled receptor activation [15,16], direct activation of adenylyl cyclases [17], and/or by inhibition of the hydrolysis of cAMP with phosphodiesterase inhibitors [18] all stimulate GLP-1 secretion. Conversely, inhibition of cAMP upon Gi-coupled receptor activation inhibits GLP-1 secretion [19,20]. Here, we showed that GPR17 inhibition of GLP-1 secretion was Gi/o-dependent, but GLP-1 secretion did not track with whole-cell cAMP. It was unexpected that the concentrations of MDL29,951 that had the strongest effects on cAMP inhibition (i.e., 100–300 nM MDL29,951) had modest effects on regulation of GLP-1 secretion, and the concentrations of MDL29,951 that yielded the most robust inhibition of GLP-1 secretion (i.e., 25 µM MDL29,951) had no significant effect on cAMP regulation. This dissociation between cAMP signaling and GLP-1 regulation was further observed in cells that were treated with high MDL29,951 concentrations alone or in conjunction with the protein kinase C activator PMA. For example, only MDL29,951-concentration-dependent increases in cAMP were observed when cells were co-stimulated with PMA (i.e., no cAMP inhibition was observed). However, significant inhibition of PMA-stimulated GLP-1 secretion was observed in response to 100 μM MDL29,951 treatment. Similar apparent dissociation of cAMP and GLP-1 secretion was also observed in response to GPBAR-A or PGE_2_ stimulation. The dissociation between whole-cell cAMP and GLP-1 secretion may be attributed to specific cAMP microdomains regulating hormone secretion. A growing body of literature suggests that GPCRs may specifically regulate diverse physiological functions that depend on distinct local pools of cAMP [47,48]. As the cAMP measurement technologies utilized in our study reported whole-cell cAMP levels, local or compartmental cAMP signaling could be obscured by a stronger whole-cell cAMP response. The complex cAMP effects may also be attributed to differential regulation of adenylyl cyclase isoforms that are uniquely modulated by G protein subunits, calcium, protein kinases, and localization in membrane microdomains [39]. Alternatively, a Gi/o-mediated signaling pathway outside of Gαi/o inhibition of cAMP signaling may inhibit GLP-1 secretion downstream of GPR17, as Gβγ subunits modulate many diverse effector pathways [49,50]. For example, voltage-dependent calcium channels (VDCCs) are capable of modulating GLP-1 secretion [51,52,53] and can be inhibited by Gβγ subunits [54]. Furthermore, mouse GPR17 expression inhibits VDCC currents in GLUTag cells [27], suggesting that hGPR17L may also inhibit GLP-1 secretion via Gi/o-Gβγ-dependent inhibition of VDCCs. Our data highlight the necessity to examine the GPCR signaling events at higher spatial resolution for regulated hormone secretion in future studies.

Our previous mouse genetic knockout studies showed that *Gpr17* ablation in key metabolic organs, including hypothalamic neurons and intestinal EECs, improve energy and glucose homeostasis [23,24,27]. In this study, we established that human GPR17 shares similar signaling properties and functions to inhibit GLP-1 secretion, further validating it as a potential target for diabetes and obesity therapy. Constitutive GPR17 activity inhibited GLP-1 secretion and MDL29,951 activation of GPR17 further inhibited F/I-stimulated GLP-1 secretion in a concentration-dependent manner. Such inhibition was mediated by Gi/o signaling and agonist-mediated inhibition was also attenuated by the small molecule GPR17 antagonist HAMI3379. A physiological context where a putative endogenous agonist has similar activity to MDL29,951 would suggest that targeting GPR17 with a neutral antagonist may lead to relatively higher GLP-1 secretion. GPR17-mediated constitutive inhibition of GLP-1 secretion and the potential for a self-activating endogenous signaling mechanism suggest that GPR17 inverse agonists may hold promise for maximizing GLP-1 secretion. HAMI3379 was reported to have modest inverse agonist properties with respect to cAMP signaling [41], but this effect was not apparent for modulation of GLP-1 secretion. A deeper and more diverse repertoire of GPR17 chemical probes is expected to enable a more thorough understanding of the pharmacological capabilities for enhancing GLP-1 secretion by targeting GPR17. Future efforts to identify and develop GPR17 antagonists and inverse agonists are expected to be fruitful for enhancing GLP-1 secretion and improving glucose homeostasis.

## 5. Conclusions

The present study tested GPR17-mediated signaling capabilities and the underlying mechanisms that contribute to regulation of GLP-1 secretion in EECs. We found that the human GPR17 long isoform inhibited GLP-1 secretion constitutively and further inhibited secretion upon activation of the receptor with MDL29,951. GPR17 effects on GLP-1 secretion were mediated by Gi/o signaling. Furthermore, the inhibition of GLP-1 secretion with MDL29,951 was concentration-dependent and was attenuated by the small molecule GPR17 nonselective antagonist HAMI3379. Our study suggests that targeting GPR17 with antagonists or inverse agonists may lead to enhanced GLP-1 secretion and could be an additional approach for treatment of metabolic diseases such as type 2 diabetes and obesity.

## Figures and Tables

**Figure 1 biomolecules-15-00009-f001:**
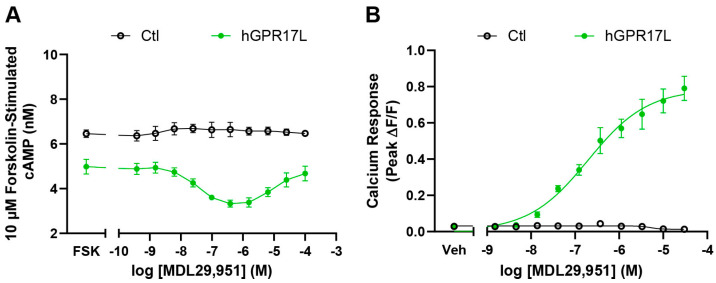
Human GPR17 long isoform modulates cAMP and calcium signaling in GLUTag cells. (**A**) Cyclic AMP accumulation was measured in GLUTag cells transduced with either control- or hGPR17L-encoding adenoviruses and stimulated with 10 µM forskolin and indicated concentrations of MDL29,951 (ranging from 0.38 nM to 100 µM). Data represent mean ± SEM of four independent experiments. (**B**) Calcium flux was measured in response to stimulation with indicated concentrations of MDL29,951 in GLUTag cells transduced with control- or hGPR17L-encoding adenoviruses. Data represent mean ± SEM of three independent experiments. Data were analyzed using two-way ANOVA with Tukey’s multiple comparisons test (see Appendix A).

**Figure 2 biomolecules-15-00009-f002:**
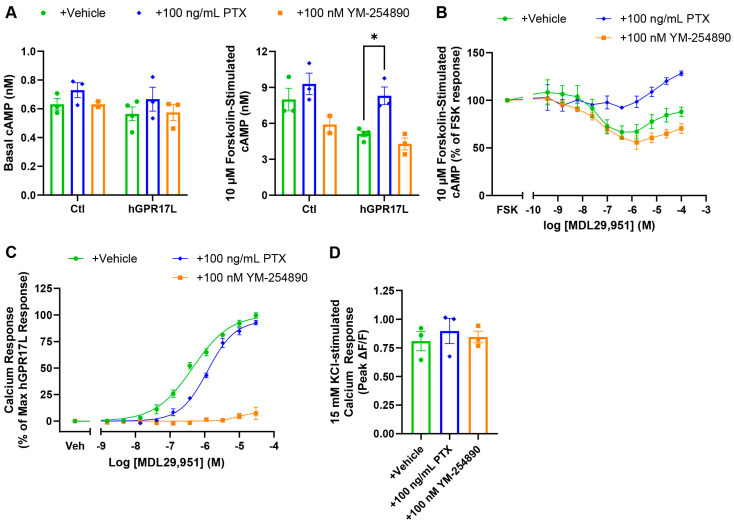
Human GPR17 long isoform regulation of cAMP and calcium signaling in GLUTag cells is mediated by Gi/o and Gq signaling. (**A**) Cyclic AMP accumulation was measured in GLUTag cells transduced with control- or hGPR17L-encoding adenoviruses when treated with vehicle, 100 ng/mL pertussis toxin (PTX), or 100 nM YM-254890 under basal conditions or upon stimulation with 10 µM forskolin. Data represent mean ± SEM of two to four independent experiments performed in triplicate and were analyzed by two-way ANOVA with Sidak’s multiple comparisons test. *, *p* < 0.05. (**B**) Cyclic AMP accumulation was measured in GLUTag cells expressing hGPR17L. Cells were stimulated with 10 µM forskolin and indicated concentrations of MDL29,951 together with vehicle, 100 ng/mL PTX, or 100 nM YM-254890. Data were expressed as a percentage of the forskolin response for each treatment condition and represent the mean ± SEM of two to four independent experiments. GLUTag cells expressing hGPR17L were treated with (**C**) MDL29,951 or (**D**) 15 mM KCl together with vehicle, 100 ng/mL PTX, or 100 nM YM-254890 and calcium mobilization was measured. Data represent mean ± SEM of three independent experiments performed in duplicate. Data in (**B**) and (**C**) were analyzed by two-way ANOVA with Tukey’s multiple comparisons test (see Appendix A). Data in (**D**) were analyzed by one-way ANOVA with Dunnett’s post hoc test compared to the vehicle treatment condition.

**Figure 3 biomolecules-15-00009-f003:**
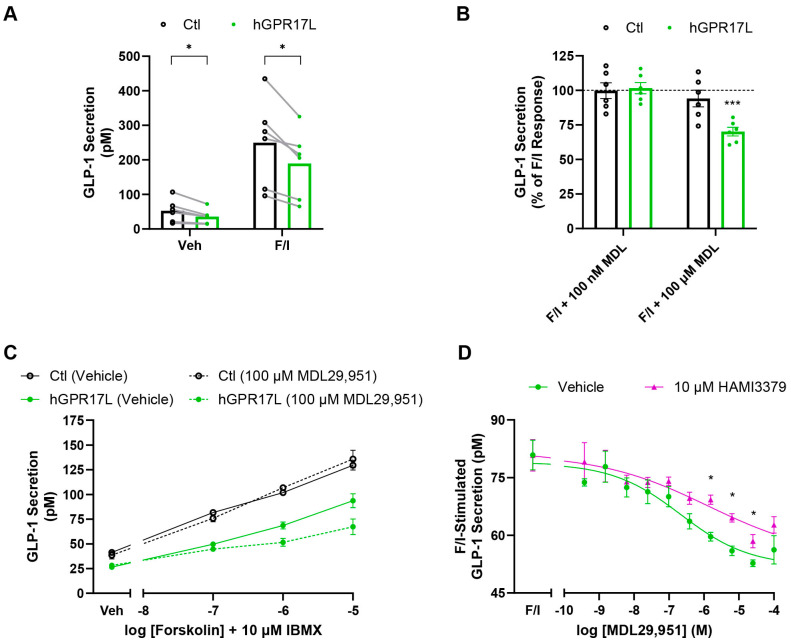
Human GPR17 long isoform negatively regulates GLP-1 secretion from GLUTag cells. (**A**) GLP-1 secretion was measured from control- or hGPR17L-expressing GLUTag cells that were treated with either vehicle or 10 µM forskolin and 10 µM IBMX (F/I). Data represent mean ± SEM of six independent experiments and were analyzed by paired t test comparing control and hGPR17L expressing cells. *, *p* < 0.05. (**B**) The effect of 100 nM MDL29,951 or 100 µM MDL29,951 treatment on F/I-stimulated GLP-1 secretion. Data represent mean ± SEM of six independent experiments and were analyzed by one sample t test compared to 100. ***, *p* < 0.001. (**C**) Effect of 100 µM MDL29,951 on GLP-1 secretion stimulated by varying concentrations of forskolin together with 10 µM IBMX. Data represent mean ± SEM of two independent experiments. (**D**) F/I-stimulated GLP-1 secretion was measured upon treatment with indicated concentrations of MDL29,951 in the presence or absence of 10 µM HAMI3379 in GLUTag cells expressing hGPR17L. Data represent four independent experiments, and statistical analysis was conducted using unpaired *t* tests between vehicle and HAMI3379-treated cells for each concentration of MDL29,951. *, *p* < 0.05.

**Figure 4 biomolecules-15-00009-f004:**
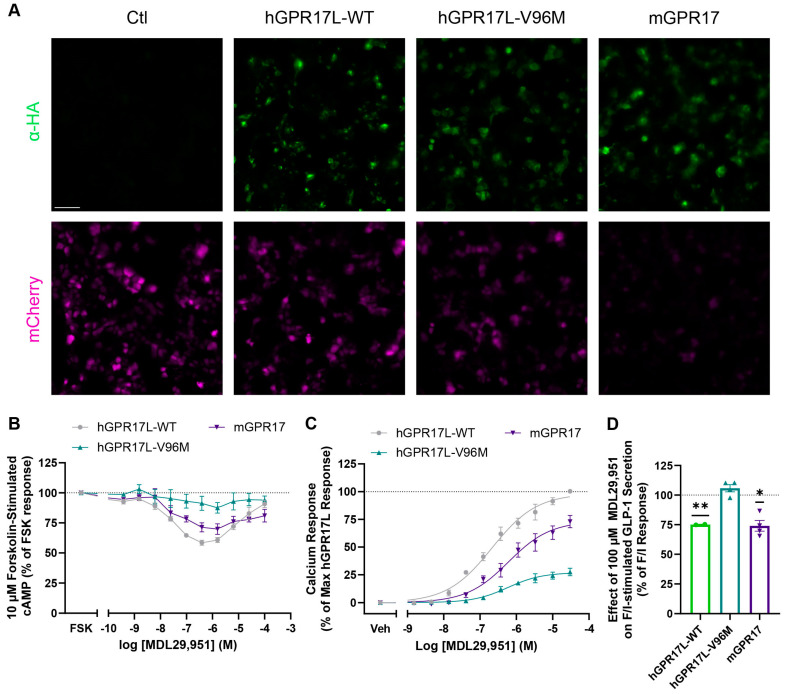
Human GPR17 long isoform-V96M and mGPR17 effects on second messenger signaling and GLP-1 secretion in GLUTag cells. GLUTag cells were transduced with control, hGPR17L-WT, hGPR17L-V96M, or mGPR17 encoding adenoviruses and were tested for GPR17 expression, second messenger signaling, and GLP-1 secretion. (**A**) Representative fluorescence images of cells that were fixed with 4% paraformaldehyde, stained with an α-HA antibody, and subsequently imaged in green and red fluorescence channels for detection of HA-tagged GPR17 and mCherry, respectively. Images are representative of four independent experiments. Scale bar, 50 µm. (**B**) Cyclic AMP was measured in cells expressing hGPR17L-V96M or mGPR17 that were treated with 10 µM forskolin and indicated concentrations of MDL29,951. (**C**) Calcium mobilization was measured in cells expressing hGPR17L-V96M or mGPR17 and treated with indicated concentrations of MDL29,951. (**D**) GLP-1 secretion was measured in response to stimulation with F/I together with 100 µM MDL29,951. Data from (**B**,**C**) represent mean ± SEM from three independent experiments for hGPR17L-V96M and mGPR17, whereas data for hGPR17L-WT are included for reference and are mean ± SEM from three experiments that were performed in parallel but included in Figure 1. Data were analyzed by two-way ANOVA with Tukey’s multiple comparisons test (see Appendix A). Data from (**D**) are expressed as a percentage of the F/I-stimulated GLP-1 secretion alone and represent mean ± SEM of two to four independent experiments and were analyzed by one-sample t test compared to 100. *, *p* < 0.05, **, *p* < 0.01.

**Figure 5 biomolecules-15-00009-f005:**
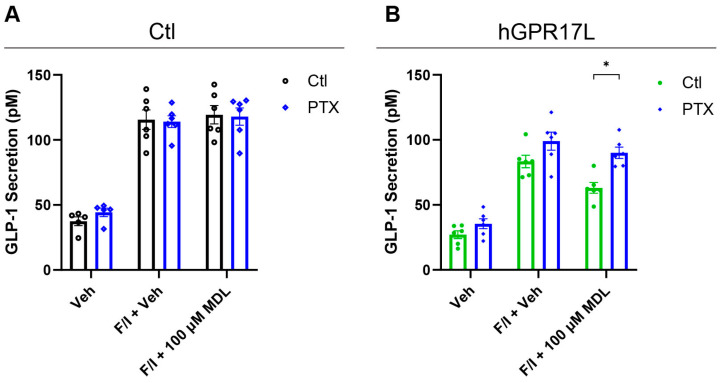
Human GPR17 long isoform regulation of GLP-1 secretion is dependent on Gi/o-mediated signaling in GLUTag cells. GLUTag cells were transduced with (**A**) control adenovirus or (**B**) adenovirus encoding hGPR17L and GLP-1 secretion was measured in cells that were treated as indicated together with either control or 100 ng/mL PTX. Data represent mean ± SEM for five or six independent experiments and were analyzed with unpaired t tests comparing matched control- and PTX-treated conditions. *, *p* < 0.05.

**Figure 6 biomolecules-15-00009-f006:**
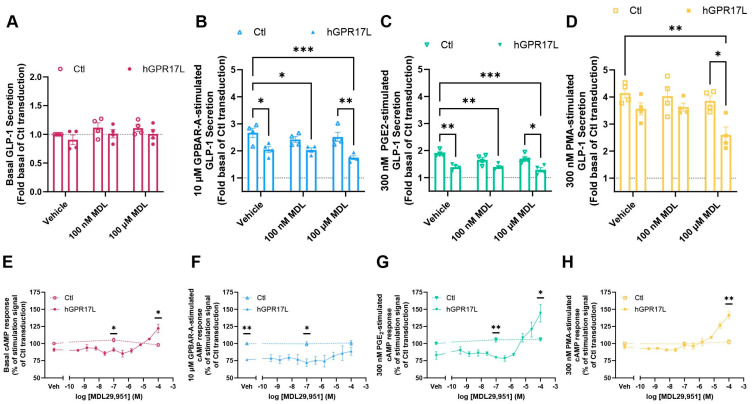
Human GPR17 long isoform distinctly regulates GLP-1 secretion and cAMP signaling in a manner that is dependent on co-stimulation conditions in GLUTag cells. GLUTag cells were transduced with control adenovirus or adenovirus encoding hGPR17L and GLP-1 secretion was measured in response to treatment with (**A**) vehicle, (**B**) 10 µM GPBAR-A, (**C**) 300 nM PGE_2_, or (**D**) 300 nM PMA together with vehicle, 100 nM MDL29,951, or 100 µM MDL29,951. Data were expressed as a fold signal of the basal treatment condition of the control transduction and represent mean ± SEM of four independent experiments performed in triplicate. Data were analyzed using one-way ANOVA with Sidak’s post hoc test. *, *p* < 0.05, **, *p* < 0.01, ***, *p* < 0.001. Cyclic AMP was measured under (**E**) vehicle, (**F**) 10 µM GPBAR-A, (**G**) 300 nM PGE2, or (**H**) 300 nM PMA stimulation conditions in combination with the indicated concentrations of MDL29,951 in control or hGPR17L-expressing GLUTag cells. Data were expressed as a percentage of the stimulation signal of the control transduction and represent mean ± SEM for two to four independent experiments performed in triplicate. Data were analyzed by unpaired t test comparing matching treatment conditions between control and hGPR17L-expressing cells. *, *p* < 0.05, **, *p* < 0.01.

**Table 1 biomolecules-15-00009-t001:** List of compounds and relevant molecular functions.

Compound	Function
MDL29,951	Synthetic GPR17 agonist
Forskolin	Adenylyl cyclase activator
3-isobutyl-1-methylxanthine (IBMX)	Phosphodiesterase inhibitor
Pertussis Toxin (PTX)	Gi/o signaling inhibitor
YM-254890	Gq signaling inhibitor
HAMI3379	Cysteinyl leukotriene receptor and GPR17 antagonist
Somatostatin (SST)	Endogenous somatostatin receptor agonist
Prostaglandin E_2_ (PGE_2_)	Endogenous prostaglandin receptor agonist
GPBAR-A	GPBAR1 (TGR5) bile acid receptor agonist
Phorbol 12-myristate 13-acetate (PMA)	Protein kinase C activator

## Data Availability

The original contributions presented in this study are included in the article and the Appendix A; further inquiries can be directed to the corresponding author.

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
