# Peer review of "G Protein-Coupled Receptor 17 Inhibits Glucagon-like Peptide-1 Secretion via a Gi/o-Dependent Mechanism in Enteroendocrine Cells"

_biomolecules, 2024, doi:10.3390/biom15010009_

Round 1

Reviewer 1 Report

Comments and Suggestions for Authors

The authors studied the GPR17-mediated signaling and the underlying mechanisms that contribute to regulation of GLP-1 secretion in Enteroendocrine Cells (EECs). This is an interesting and well written manuscript, addressing an important problem of optimizing GLP1 signaling in the context necessary for metabolic homeostasis. For clarity, I strongly recommend adding the abbreviation list at the beginning of the text. Also, please change the scale in Fig.6A, the bar graphs are not visible in a current version. 

Reviewer 2 Report

Comments and Suggestions for Authors

The present study was designed to investigate the GPR17 signalling mechanisms that regulate GLP-1 secretion. The results obtained are clear and convincing and offer the possibility of opening new avenues for understanding the system.

Author Response

We thank the reviewer for taking the time to review this manuscript. We appreciate the supportive review!

Reviewer 3 Report

Comments and Suggestions for Authors

The current manuscript by Conley et al. used pharmacological probes to investigate the signaling mechanisms by which GPR17 regulates GLP-1 secretion in EEC. Overall, this study was done in a physiologically relevant cell line. The major findings include the regulation of GLP-1 secretion by hGPCR17L depends on a Gαi-mediated but not Gq-mediated signaling. This study showed a biphasic pattern of cAMP modulation by hGPCR7L in response to a synthetic agonist MDL treatment. However, the reversal of cAMP inhibition by higher concentrations of MDL does not associate with GLP-1 secretion.

The manuscript is well-prepared, and the results are appropriately interpreted to support the major conclusions. There are several questions to be clarified.

1.        What are the endogenous expression levels of GPR17 in the GLUTag cells (mRNA and protein)? How would the endogenous GPR17 protein if any impact the interpretation of the data?

2.        Fig 1A: what are the possible mediators that reverse the cAMP inhibition by the higher concentration of MDL?

3.        Fig 3D: the attenuation of agonist-stimulated GLP-1 secretion by GPR17 antagonist is less robust than the PTX inhibition of Gi signaling in Fig 2C.  What is the possible explanation for these differences?

4.        Fig 4A: why mCherry signaling is much lower in the mGPR17 panel than in the other groups?

5.        Fig S5: the data in Fig S5 A-B and G-H only include two data points in each group. It will be necessary to include more data points to ensure rigor and support the conclusion.

6.        The author discusses several possible mechanisms that lead to the dissociation between cAMP and GLP-1 secretion. It would be more informative to provide more data or literature to probe or explain the possible regulators involved, such as if there is any possible cross-talk between the Gq and Gi-mediated signaling events. 
